# Design, Modeling, and Testing—A Compact Variable-Stiffness Actuator for Knee Joint Dimensions

**DOI:** 10.3390/mi16121365

**Published:** 2025-11-29

**Authors:** Guoning Si, Zilong Guo, Zhuo Zhang

**Affiliations:** 1School of Health Science and Engineering, University of Shanghai for Science and Technology, Shanghai 200093, China; 232322307@st.usst.edu.cn; 2School of Modern Posts Industry, Xi’an University of Posts and Telecommunications, Xi’an 710121, China; zhangzhuozz@xupt.edu.cn

**Keywords:** variable-stiffness mechanism, knee exoskeleton, exoskeleton design, hemiparetic gait rehabilitation

## Abstract

Stroke rehabilitation exoskeletons require joint mechanisms capable of replicating physiological stiffness modulation to adapt to varying gait phases. This paper presents a novel compact variable-stiffness mechanism (VSM) for knee exoskeletons, based on a simplified three-bar linkage topology. The proposed design achieves a pre-configurable quasi-stiffness range of 0.15–2.0 NM/deg. Static characterization under a 2 kg load demonstrated up to 23.0 N of collision force attenuation in softening regimes (λ < 2.3) through passive viscoelastic dissipation, whereas hardening behavior (λ ≥ 2.3) preserved precise torque-angle characteristics scalable to physiological loading. A parametric analysis showed an 89% correlation between the theoretical and scaled experimental stiffness profiles for values from 0.5 to 2.5. The proposed architecture enables decoupled optimization of impact safety and positional precision, offering a clinically adaptable solution for hemiparetic gait assistance.

## 1. Introduction

Accelerated population aging and urbanization have precipitated a discernible escalation in the prevalence of stroke risk factors, and concomitantly led to a marked increase in the burden of stroke-related disability, particularly in China [1]. Post-stroke hemiplegia and gait abnormalities constitute primary motor dysfunctions among stroke survivors, representing predominant determinants adversely impacting activities of daily living and quality of life [2,3]. Post-stroke rehabilitation is a progressive, dynamic process aimed at restoring physical, cognitive, motional, communicative, and social functions. Substantial clinical evidence demonstrates that integrating physical rehabilitation with task-oriented functional training can greatly enhance patients’ post-stroke functional recovery [4]. For non-ambulatory patients, robot-assisted gait training combined with physical therapy has been shown to enhance the probability of regaining independent ambulation [5].

Over the past decade, numerous robotic systems and control strategies for gait rehabilitation have been developed [6]. In clinical rehabilitation settings, treadmill-based exoskeletons, such as PeXO [7], MRR [8], LOPES [9], and C-Mill [10] systems, enable repetitive, task-specific locomotor training during the early recovery phase. For community-based or home rehabilitation, wearable exoskeletons must be lightweight, safe, and easy to take on and off, while providing effective gait assistance.

Many portable lower limbs have been proposed [11,12,13,14,15]; however, most employ fixed-stiffness joints, which limit adaptability due to inter-subject variability in gait speed and joint stiffness [16,17]. This limitation highlights the importance of implementing variable-stiffness mechanisms (VSMs), which allow mechanical adaptation to different gait phases. In general, VSMs achieve stiffness modulation by altering the characteristics of springs—such as effective length or pretension force [18]. Bio-inspiration underpins designs emulating the antagonistic action of musculoskeletal systems, incorporating spring pretension adjustment to modify stiffness settings [19]. Xie et al. developed a compact elbow exosuit driven by pneumatic artificial muscles (PAMs) that provides effective assistance for elbow flexion–extension movements. This design highlights the potential of PAMs as lightweight and biomimetic actuators for joint actuation in portable assistive systems [20]. Subsequent studies achieved concurrent joint positioning and stiffness modulation via independent motors, thereby reducing actuator footprint and mass. Notable implementations include the MINDWALKER [21], MACCEPA 2.0 [22], VS-joint [23], FSJ [24], SoftJoint I [25], and LC-VSM [26] architectures. Recently, Li et al. [27] introduced a modular, reconfigurable VSM based on a four-bar linkage, and Bowden cable actuation, enabling stiffness adjustment through changes to the spring preload. Building upon this foundation, this study proposes a compact VSM specifically optimized for knee exoskeleton applications. Based on a three-bar linkage mechanism, it employs the ratio of output-to-input link lengths as the key kinematic variable for stiffness adjustment. By modulating this link ratio parameter, the mechanism can switch between softening and stiffening behavior modes, enabling phase-adaptive impedance matching with biological knee dynamics throughout the gait cycle. Compared to fixed-stiffness actuators, this approach offers significant advantages: the softening mode enhances shock absorption during external impacts, improving safety in human–machine interactions, while the stiffening mode increases gait-cycle stability and reduces metabolic expenditure. Furthermore, the lever ratio-based control scheme decouples stiffness adjustment from position control, simplifying stiffness variation implementation while maintaining mechanical simplicity. The key contributions of this study include the following: (i) a novel mechanical design tailored to joint dimensions and biomechanical constraints; (ii) a mathematical model defining input–output lever ratio constraints and stiffness boundaries; (iii) static and dynamic characterization of the VSM prototype; (iv) modularization of the VSM for seamless integration into knee exoskeleton systems for gait rehabilitation in hemiplegic patients.

The remainder of the paper is organized as follows. Section 2 reviews the relevant literature, introduces the link length ratio parameter, and details the mechanical design, modeling, and prototype implementation. Section 3 reports experimental results from static testing. Section 4 discusses the findings and Section 5 concludes the paper.

## 2. Mechanical Design of VSM

### 2.1. Basic Principles of Stiffness Variation

The variable-stiffness mechanism (VSM) proposed in this paper builds on the linkage concept of Li and Bai [27]. In this study, the completeness of this concept is briefly reiterated and the parameters are optimized for subsequent mechanical design. Although the mechanism is topologically a four-bar linkage, in our specific configuration the ground link is reduced to zero-length, causing the input and output links to share a common rotational axis. This special-case geometry allows the system to be treated as a three-link structure for analytical convenience, without altering its underlying kinematic class. Such a simplification is consistent with the principle that, in a four-bar linkage, when the ground link becomes zero-length, the speed and torque ratios approach unity and the mechanism exhibits singular behavior—a condition previously noted in ref. [27]. As illustrated in Figure 1a,b, rod-1 (input link) and rod-3 (output link) are mounted to the base via revolute joints, while the elastic rod-2 (a spring-coupled element) connects the distal ends of rod-1 and rod-3. The spring-coupled unit provides a pre-configurable viscoelastic path between the two links, enabling stiffness modulation without adding bulk actuators to the joint. The output torque of the mechanism is generated by the elastic force applied along rod-2. Because the spring can only be pulled along its own axis, the force transmitted to the linkage is directed along the spring line. With respect to the output joint, only the tangential part of this force produces rotation; the part that points through the joint center merely compresses or extends the mechanism without turning it. The resulting torque therefore depends on two geometric factors that change with configuration: (i) the direction of the spring force relative to the joint; and (ii) the effective lever arm, i.e., the perpendicular distance from the joint center to the spring’s line of action. When the spring line passes close to the joint center, the lever arm becomes small and the torque tends to vanish; when the line is farther away and more transverse, the torque increases. As the joint angle and the attachment ratio change, these two quantities are variable. The torque $\tau_{out}$ generated at the output link of $l_{3}$ arises from the elastic force $F_{S}$ in the spring-coupled unit, transmitted through the linkage Jacobian J:(1)$$\tau_{\mathrm{out}}=F_{s}\cdot J$$
with(2)$$J=\left|{\hat{l}}_{2}\times \vec{l_{1}}\right|=\frac{\partial l_{2}}{\partial \theta }$$

The Jacobian J characterizes the instantaneous rate of change of the spring length $l_{2}$ with respect to the angular displacement θ between links $l_{1}$ and $l_{3}$. Here, $\vec{l_{1}}$ denotes the position vector from the pivot point to the spring attachment point (with magnitude $l_{1}$), and ${\hat{l}}_{2}$ represents the unit direction vector along the spring axis. The term $\partial l_{2}$ corresponds to the differential change in spring length due to angular variation ∂θ. Furthermore, the Jacobian J can be expressed in the following form:(3)$$J=\left|{\hat{l}}_{2}\times \vec{l_{1}}\right|=\frac{\left|{\vec{l}}_{2}\times \vec{l_{1}}\right|}{l_{2}}=\frac{l_{1}l_{3}\sin\theta }{l_{2}}=\frac{l_{1}l_{3}\sin\theta }{\sqrt{l_{1}^{2}+l_{3}^{2}-2l_{1}l_{3}\cos\theta }}$$

$F_{S}$ is provided by the $l_{2}$ spring-coupled element, satisfying Hooke’s law:(4)$$F_{S}=k\cdot \left(l_{2}-l_{2.0}\right)+F_{0}=k\cdot \Delta l_{2}+F_{0}$$
where $l_{2.0}$ is the original length of $l_{2}$ when the angle θ between $l_{1}$ and $l_{3}$ is 0, $\Delta l_{2}$ is the spring change amount, and $F_{0}$ is the pretension of the elastic element when θ=0.

Joint stiffness is related to θ, which is derived from the differential of Equation (1):(5)$$K_{\mathrm{eq}}=\frac{\delta \tau_{\mathrm{out}}}{\delta \theta }=J\cdot \frac{\delta F_{S}}{\delta \theta }+\frac{\delta J}{\delta \theta }\cdot F_{S}=J\cdot k\cdot J+\frac{\delta J}{\delta \theta }\cdot F_{S}=J^{2}k+\frac{\delta J}{\delta \theta }\cdot F_{S}$$

Substituting Equation (4) into Equation (5) and rearranging it, we can obtain the relation of the equivalent stiffness $K_{eq}$ to k and the spring preload $F_{0}$:(6)$$K_{\mathrm{eq}}=J^{2}k+\frac{\delta J}{\delta \theta }\cdot F_{S}=J^{2}k+\frac{\delta J}{\delta \theta }\cdot k\cdot \Delta l_{2}+\frac{\delta J}{\delta \theta }\cdot F_{0}$$

We can observe that the equivalent stiffness $K_{eq}$ in Equation (6) is mathematically partitioned into two distinct components, which are determined by the spring coupling mechanism k and the preload force $F_{0}$ provided by the spring.

For convenience of study, we introduce the ratio of the input rod length to the output rod length $\lambda =l_{3}/l_{1}$ and specify that $l_{3}$ is adjustable; then, $l_{2}=l_{1}\sqrt{1+\lambda^{2}-2\lambda \cos\theta }$, and we rearrange Equation (3) to obtain(7)$$J=\frac{\lambda l_{1}\sin\theta }{\sqrt{1+\lambda^{2}-2\lambda \cos\theta }}$$

Differentiating Equation (7) with respect to θ and substituting $l_{2}=l_{1}\sqrt{1+\lambda^{2}-2\lambda \cos\theta }$ yields(8)$$\begin{array}{c} \frac{\delta J}{\delta \theta }=\frac{\lambda l_{1}^{2}\cos\theta }{l_{1}\sqrt{1+\lambda^{2}-2\lambda \cos\theta }}-\frac{\lambda^{2}l_{1}^{4}\sin^{2}\theta }{l_{1}^{3}{\left(1+\lambda^{2}-2\lambda \cos\theta \right)}^{3/2}}\\ =\frac{\lambda l_{1}\left[\cos\theta (1+\lambda^{2}-2\lambda \cos\theta )-\lambda \sin^{2}\theta \right]}{{\left(1+\lambda^{2}-2\lambda \cos\theta \right)}^{3/2}} \end{array}$$

When the angle θ between $l_{1}$ and $l_{3}$ is 0, the initial length $l_{2.0}$ of the spring coupling unit is(9)$$l_{2.0}=l_{1}\sqrt{1+\lambda^{2}-2\lambda }=l_{1}|1-\lambda |=\left\{\begin{array}{ll} l_{1}(1-\lambda ), & l_{3}<l_{1}\\ 0, & l_{3}=l_{1}\\ l_{1}(\lambda -1), & l_{3}>l_{1} \end{array}\right.$$(10)$$\Delta l_{2}=l_{2}-l_{2.0}=l_{1}\left(\sqrt{1+\lambda^{2}-2\lambda \cos\theta }-|1-\lambda |\right)$$

Note that $D=\sqrt{1+\lambda^{2}-2\lambda cos\theta }$. By substituting Equations (6)–(10) into the general stiffness expression and simplifying, we obtain the final form of the stiffness Equation (11):(11)$$K_{\mathrm{eq}}=kl_{1}^{2}\cdot \frac{\lambda }{D^{3}}\left\{\lambda D\sin^{2}\theta \right.+\left[\cos\theta (1+\lambda^{2})-\lambda \cos^{2}\theta -\lambda \right]\times \left.\left(D-|1-\lambda |\right)\right\}$$

### 2.2. Analysis of the Proposed VSM

Parametric optimization of the variable-stiffness actuator (VSA) requires a comprehensive analysis of its characteristics to establish design constraints. By strategically simplifying the equations under specific kinematic conditions—specifically, an initial inter-link angle of θ=0 and the length configuration where $l_{3}=l_{1}+l_{2}$—the equivalent stiffness $K_{\mathrm{eq}}$ in Equation (6) becomes analytically decoupled from the spring preload force $F_{0}$:(12)$$K_{\mathrm{eq}}=J^{2}k+\frac{\delta J}{\delta \theta }\cdot F_{S}=J^{2}k+\frac{\delta J}{\delta \theta }\cdot k\cdot \Delta l_{2}$$

This essential simplification occurs because the configuration $l_{3}=l_{1}+l_{2}$ negates the effects of spring deformation, thereby eliminating terms dependent on $F_{0}$ in the stiffness expression.

When $l_{3}<l_{1}+l_{2}$, in order to keep the preload $F_{0}=0$, we do not consider the contribution of the compression spring in the spring coupling unit. Figure 2 shows that the initial angle θ between $l_{3}$ and $l_{1}$ is not 0:(13)$$\tau_{\mathrm{out}}=J\cdot F=\frac{l_{1}l_{3}\sin\theta^{*}}{l_{2}}\cdot k\Delta l_{2}$$
where $\theta^{*}=\theta_{0}+\theta_{rel}$. Here, $\theta_{0}$ denotes the initial angle between link $l_{3}$ and $l_{1}$, and $\theta_{rel}$ represents the relative angular displacement induced during the actuation of $l_{2}$. The angle $\theta_{0}$ can be derived as follows:(14)$$l_{2.0}=\sqrt{l_{1}^{2}+l_{3}^{2}+2l_{1}l_{3}\cos\theta_{0}}$$(15)$$\theta_{0}=\cos^{-1}\left(\frac{l_{2.0}^{2}-l_{1}^{2}-l_{3}^{2}}{2l_{1}l_{3}}\right)$$

Conversely, when $l_{3}>l_{1}+l_{2}$, a non-zero preload force $F_{0}\neq 0$ is applied, with an initial angle θ=0 for $l_{3}$ and $l_{1}$, and a stiffness the same as in Equation (6).

Critically, Equation (11) demonstrates that the equivalent stiffness $K_{eq}$ is exclusively determined by the parameters λ and θ. This section systematically investigates how these parameters shape the stiffness characteristics. As illustrated in Figure 3a, the output torque of the VSM increases monotonically with rising values of λ as well as θ. Figure 3b further partitions the $K_{eq}$ domain into three distinct regimes, corresponding to hardening behavior, softening behavior, and negative stiffness. Specifically, joint stiffening refers to the enhancement of equivalent stiffness under deformation, whereas softening denotes a reduction in stiffness responsiveness.

A parametric analysis indicates that λ serves as the principal regulator of joint stiffness properties. By strategically tuning $\lambda =l_{3}/l_{1}$, a variety of stiffness regimes can be engineered. For our analysis, $\lambda =l_{3}/l_{1}$—which has an initial value of 2.3 and is adjustable at setup up to 2.5—serves as the principal parameter for regulating joint stiffness. A reduced λ compresses the range of softening behavior but simultaneously elevates $K_{eq}$ at low θ. When the condition $l_{3}>l_{1}+l_{2}$ is satisfied, the system stiffness formulation additionally incorporates a preload force, as reflected in Equation (6). Notably, Figure 3 shows a pronounced increase in both joint torque and stiffness once λ>2.3.

Moreover, the configuration $l_{3}=l_{1}+l_{2}$ induces a spring-dominant mode of stiffness modulation, as formalized in Equation (12) and Equation (13). Figure 4 provides a quantitative depiction of the coupled evolution of torque and stiffness with respect to θ and k. As illustrated in Figure 4, the system torque $\tau_{out}$ exhibits a pronounced positive correlation with both the angle θ and the spring constant k. Furthermore, the rates of change of both the output torque and the system stiffness $K_{eq}$ increase significantly with higher values of k, indicating enhanced sensitivity of the system’s stiffness response under greater spring constants.

### 2.3. Structure Design and Prototype

The physical prototype, shown in Figure 6, integrates three primary subsystems: a drive module, the variable-stiffness mechanism, and an output measurement module.

As illustrated in Figure 5a, when $l_{3}<l_{1}+l_{2}$, a zero-preload initial angle is established between the input and output rods, yielding a zero-preload condition within the spring unit—consistent with the analytical formulation in Equation (13). Under conditions where $l_{3}=l_{1}+l_{2}$ or $l_{3}>l_{1}+l_{2}$, the three-bar linkage assumes a collinear configuration, with a preload force being introduced exclusively when $l_{3}>l_{1}+l_{2}$. Figure 5b further illustrates the operational states of the VSM under varying deflection angles for n=3 and λ=2.5, where spring elongation progressively generates output torque.

The drive module consists of a position-controlled servo motor, which is realized as an integrated electromechanical assembly combining a brushless DC motor, a harmonic reducer, an optical encoder, and dedicated servo-drive electronics. The actuator is mounted onto the experimental platform via precision-machined brackets, providing controlled torque and positional input to the VSM. Command signals are regulated through a host computer, while rigid coupling interfaces at both termini ensure precise kinematic alignment among all subsystems.

Leveraging the previously established principle of variable stiffness modulation, a reconfigurable VSM was developed comprising three functional stages. As shown in the enlarged view in Figure 6, the input stage employs planetary reduction gearing, with three radially arranged linkages serving as input rods. The spring-coupling stage directly connects the input and output shafts, while the output stage incorporates an adjustable T-nut that enables manual modulation of link length. The spring characteristics can be further tailored either by substituting elastic elements of varying stiffness or by reducing the number of parallel springs coupled between the shafts. Manual adjustment of the T-nut pivot enables operational mode transitions between softening stiffness regimes ($\partial K_{eq}/\partial \theta <0$) and stiffening regimes ($\partial K_{eq}/\partial \theta >0$). Notably, these adjustments are made manually during the initial setup phase and remain fixed throughout operation. The current implementation does not support real-time stiffness modulation, but rather pre-configured tuning tailored to specific use-case scenarios. The principal design parameters of the prototype are summarized in Table 1.

An experimental platform was constructed to evaluate the VSM’s performance in terms of stiffness identification, collision safety assessment, and trajectory-tracking validation. The testbed configuration is depicted in Figure 7. A linearized automatic control scheme was implemented to regulate motor rotation angles, serving as a biomechanically validated alternative to conventional manual leg oscillation protocols. The VSM prototype’s input shaft is driven by a 57 mm frame brushless DC motor (Dinotec DT57BL70–230) through a Joyland PLF60 proportional planetary gearbox (20:1 reduction ratio). A reaction pendulum, mechanically linked to the output shaft via an HLT-151 torque transducer (±0.01 Nm accuracy), enables controlled torque measurement. Angular displacement θ is measured with 0.5° resolution using an Oid-Encoder OID-R3806D absolute encoder mounted on the pendulum axis. All actuators and sensors are interfaced with a host computer through shielded communication cables, allowing centralized monitoring and control via an integrated human–machine interface (HMI).

Through systematic parameter optimization on the developed platform, the spring configuration cataloged in Table 2 was selected.

## 3. Experimental Evaluation

To experimentally validate these findings, quasi-static experiments were conducted to characterize the torque-deflection behavior of the mechanism prototype. Five experimental cases were considered, as illustrated in Figure 5a. Physical implementation of each case was achieved by manually adjusting the T-nut on output rod $l_{3}$. Three cases are primarily discussed: Cases 1, 2, and 3 correspond to $l_{3}<l_{1}+l_{2}$ scenarios, while cases 4 and 5 correspond to collinear configurations of $l_{1}$, $l_{2}$, and $l_{3}$. Specifically, case 4 satisfies $l_{3}=l_{1}+l_{2}$, whereas case 5 satisfies $l_{3}>l_{1}+l_{2}$, resulting in a preload condition within the spring. For each trial, torque was applied either by offsetting the output shaft of the pendulum prototype or by direct actuation from the input motor. The torque and deflection data were simultaneously acquired through integrated sensors.

The experimental results are comprehensively summarized in the accompanying figures. Figure 8a illustrates the torque reflection relationship for a configuration with n=3 and k= 2.5 N/mm across different rod-length ratios. Similarly, Figure 8b depicts the corresponding torque reflection curves under altered parameters (n=3, k= 4.3 N/mm), further validating the reliability of the proposed mechanism. In Figure 8a,b, the theoretical and experimental results show strong agreement, validating the torque prediction model. Comparative analyses between theoretical stiffness predictions and experimentally derived stiffness values, the latter obtained via post-processing of measured data, are presented in Figure 9. For each λ, the profile exhibits distinct trends: cases with smaller λ = {0.7,1.0,1.5} exhibit softening behavior ($\partial K_{eq}/\partial \theta <0$), while larger λ = {2.3,2.5} yield stiffening behavior ($\partial K_{eq}/\partial \theta >0$). This confirms the mechanism’s ability to generate both positive and negative stiffness slopes by simply reconfiguring geometric parameters at setup. The excellent agreement between theory and experiment demonstrates the feasibility and repeatability of such programmable stiffness profiles. Finally, oscillation tests were performed by applying precisely controlled torques to the oscillating components; representative results from these tests are shown in Figure 10. In Figure 10a,b, the monotonic torque–angle relationship confirms stable system behavior, while differences among the curves highlight how spring stiffness k and geometric parameter λ jointly shape the dynamic stiffness characteristics of the mechanism. These figures collectively confirm that the proposed mechanism can realize both positive and negative stiffness variation trends by adjustment of λ, a key characteristic of variable-stiffness mechanisms.

## 4. Discussion

The experimental characterization of the VSM successfully validated its core mechanical principles and performance. The close agreement between the theoretical predictions and measured torque–displacement data (Figure 8), with a maximum error of ±56 N ·mm, confirms the accuracy of the established mathematical model for the three-link, lever-based design.

The parametric analysis revealed that the rod-length ratio (λ) is the dominant factor in shaping the mechanism’s stiffness behavior. Specifically,

Low-λ regimes ($l_{3}<l_{1}+l_{2}$) produced a softening stiffness profile, characterized by high positional accuracy (error <2° under loads of 5 Nm). This regime also demonstrated effective passive energy dissipation, absorbing up to 0.8 J during impact events, which is a critical feature for operational safety.High-λ regimes ($l_{3}\geq l_{1}+l_{2}$) induced a pronounced hardening stiffness behavior, where joint stiffness increased monotonically with deflection. This design enables the joint to provide substantial mechanical resistance under high loads.

The mechanism’s dynamic performance was further quantified through swing testing. As shown in Figure 10, experimental hysteresis loops closely conformed to theoretical predictions, despite peak deviations of ±220 N · mm, caused by backlash in the planetary gearset. The tests confirmed that increasing the spring preload force (F_0_) effectively amplifies the intensity of the hardening behavior, generating high stiffness gradients, up to 28.5 N⋅m/deg. These results confirm that the proposed mechanism can realize both softening and hardening behaviors depending on the configuration, offering nonlinear stiffness modulation suited for diverse functional needs.

Despite the strong overall performance, the analysis identified two key mechanical limitations:Transient torque oscillations: At the initial 5° of angular displacement, peak errors reached 11%, primarily attributable to static friction breakaway at the shaft-housing interface.Sustained motion error: Beyond the initial transient, a smaller sustained error (~8%) remained, largely caused by stick–slip vibrations during continuous motion.

In stroke rehabilitation, matching joint stiffness to the patient’s condition and recovery stage is essential. The proposed mechanism supports this by enabling stiffness parameters—such as the rod-length ratio and spring preload—to be adjusted during setup, prior to operation. For example, in early-stage recovery, a lower preset stiffness can facilitate compliant joint behavior, reducing the mechanical load on weakened limbs and minimizing the risk of secondary injury during passive or therapist-assisted movements. In contrast, higher stiffness configurations enhance joint stability in later phases, where patients engage in weight-bearing tasks like supported standing or ambulation.

While the stiffness is not modulated in real time, the mechanism’s ability to switch between softening and hardening regimes through simple structural reconfiguration allows for efficient adaptation to different rehabilitation scenarios. This makes it particularly suitable for modular assistive devices and semi-passive orthoses that require task-specific stiffness customization.

Additionally, the structure’s passive energy-dissipation capability—useful for absorbing unexpected loads such as those caused by spastic reflexes—improves collision safety. The absence of bulky actuators also supports low-power operation, making the design favorable for wearable, battery-powered rehabilitation systems. Beyond the above application potential, it is important to note that this study is still situated within an early-stage development phase, which presents broader scope limitations. Most notably, the current work has not yet incorporated automation into the testing procedure, lacks full system integration with a human-in-the-loop controller, and has not been validated through large-scale clinical trials. To address these aspects, our immediate research roadmap will focus on the following:(1)Developing an automated testing framework to enhance protocol consistency and efficiency;(2)Implementing and validating a robust control system that integrates real-time human input;(3)Initiating pilot clinical studies to benchmark performance against established physiological benchmarks.

In summary, this investigation presents a novel structural design for a compact, variable-stiffness exoskeletal joint. Its principal contribution lies in a compliant joint architecture that enables significant stiffness modulation through straightforward manual reconfiguration of the rod-length ratio (λ) and spring constant (k). The ability to switch between softening, linear, and stiffening regimes makes the mechanism highly adaptable for a wide range of operational requirements in robotic systems. Future work will focus on implementing automated parameter adjustment and mitigating transient friction effects through optimized surface coatings and bearing selections.

## 5. Conclusions

This study established a biomimetic variable-stiffness joint paradigm derived from a simplified three-bar mechanism, with a rod-length ratio λ adjustable at setup. Experimental and theoretical analyses confirmed that the proposed structure enables distinct softening and stiffening behaviors, thereby validating its feasibility as a variable-stiffness actuator. The novel VSM functions effectively either as a passive actuator or a flexible drive unit when coupled with electric motors, making it directly applicable to rehabilitation and assistive robotics, particularly for knee orthoses. Importantly, by allowing stiffness configuration during setup—via changes in rod-length ratio or spring preload—the mechanism enables the joint to be tailored for different therapy stages. In early recovery, a low stiffness setting supports compliant, low-resistance movement to minimize strain on impaired limbs, while in later phases, higher stiffness enhances joint stability for tasks such as standing or gait training. These preset stiffness modes allow safe, progressive rehabilitation without requiring active-control hardware. Beyond validating the stiffness modulation performance, the results highlight the potential of the VSM to enhance joint stability, collision safety, and energy efficiency in human–robot interaction. Future work will focus on case studies for modular robotic joints, targeting quantitative knee stiffness requirements during ambulation and emphasizing phase-dependent stiffness adaptation across gait-cycle transitions.

## Figures and Tables

**Figure 1 micromachines-16-01365-f001:**
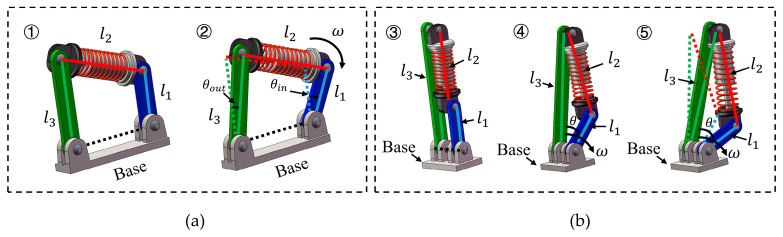
Working principle of the VSM. (**a**) Prototype of four-bar linkage. (**b**) Simplification of the initial state and stretched state of the three-link mechanism.

**Figure 2 micromachines-16-01365-f002:**
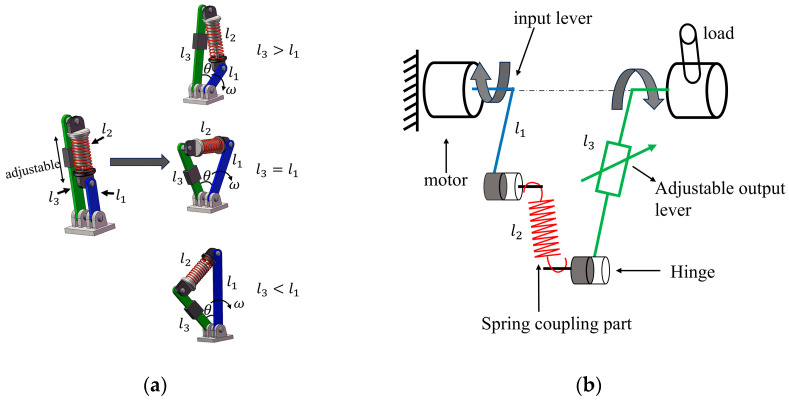
Construction of adjustable VSM and VSA. (**a**) Compliant mechanism of nonlinear stiffness behavior with different rod-length ratios. (**b**) Adjustable nonlinear-stiffness-behavior actuator corresponding to VSM.

**Figure 3 micromachines-16-01365-f003:**
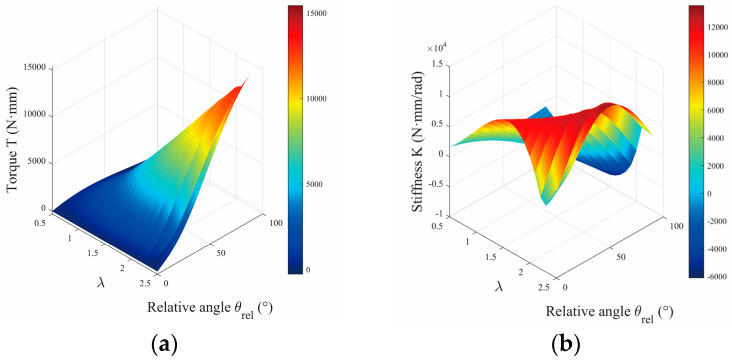
(**a**) Influence of deflection angle $\theta_{rel}$ and different rod-length ratios λ on VSM output torque $\tau_{out}$. (**b**) Stiffness performance $K_{eq}$ affected by $\theta_{rel}$ and λ, with n=3 and λ=2.3.

**Figure 4 micromachines-16-01365-f004:**
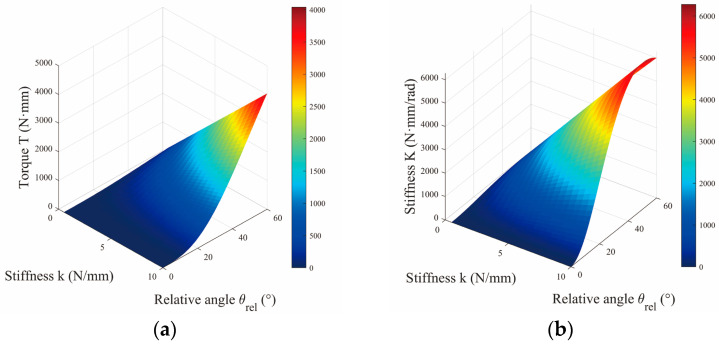
(**a**) Influence of deflection angle $\theta_{rel}$ and different stiffnesses k on VSM output torque $\tau_{out}$. (**b**) Stiffness performance $K_{eq}$ affected by $\theta_{rel}$ and λ, with n=3 and λ=2.3.

**Figure 5 micromachines-16-01365-f005:**
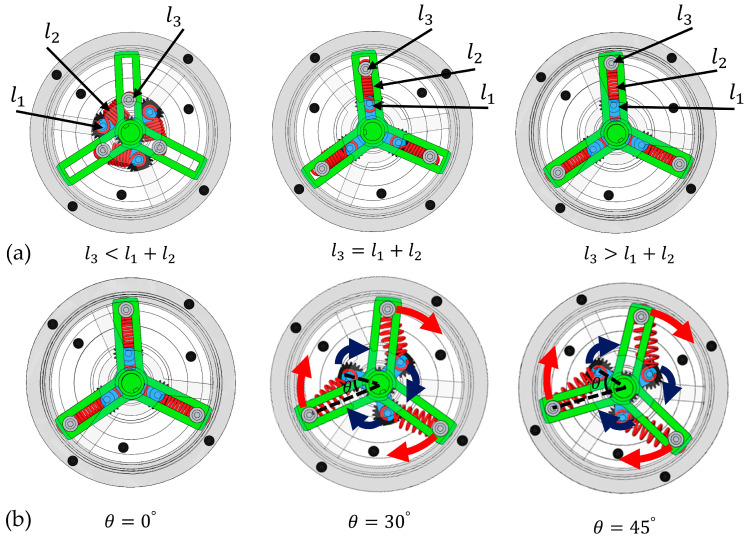
(**a**) Initial state diagram of the proposed VSM under different rod-length ratios λ, and (**b**) state diagram of VSM driven by different deflections when n=3 and λ=2.5.

**Figure 6 micromachines-16-01365-f006:**
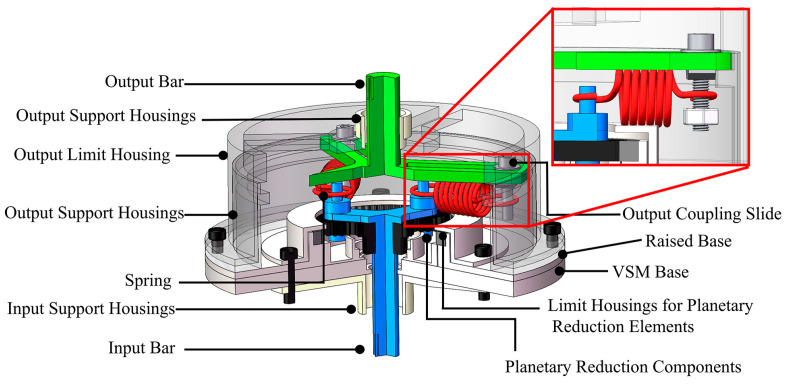
CAD model of the compliant actuator designed with the VSM.

**Figure 7 micromachines-16-01365-f007:**
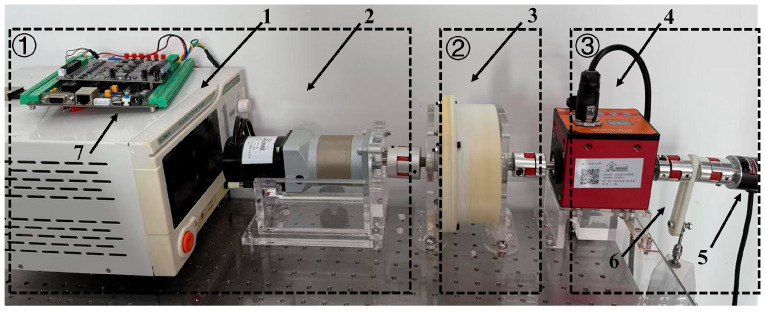
VSM test rig constructed with (1) power supply unit, (2) motor gearbox, (3) VSM, (4) torque sensor, (5) encoder, (6) load cell, and (7) PC.

**Figure 8 micromachines-16-01365-f008:**
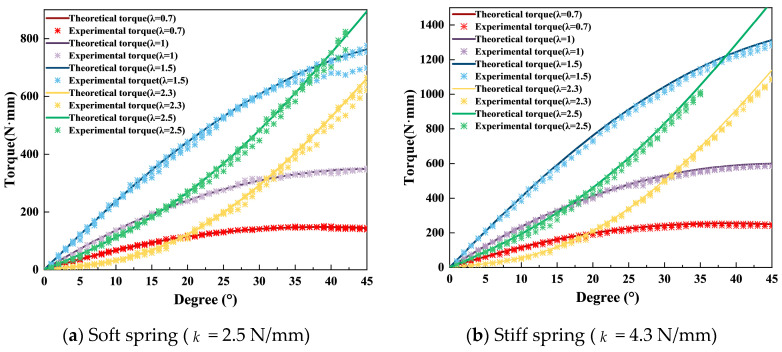
(**a**) Quasi-static torque–deflection curves for a configuration with *n* = 3 and spring stiffness *k* = 2.5 N/mm evaluated across multiple rod-length ratios *λ* = {0.7,1.0,1.5,2.3,2.5}. Solid lines represent theoretical predictions, while markers indicate experimental measurements. A larger *λ* yields a steeper torque–angle slope. (**b**) Same protocol with a stiffer spring (*k* = 4.3 N/mm). Relative to (**a**), both the torque level and the local slope increase across all *λ*, reflecting the effect of spring stiffness; agreement between theory and experiment is maintained.

**Figure 9 micromachines-16-01365-f009:**
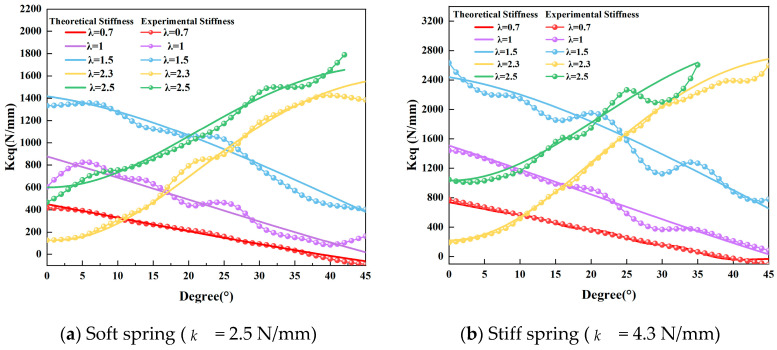
(**a**) Comparison of theoretical stiffness and experimentally characterized stiffness as a function of deflection angle θ, corresponding to different rod-length ratios λ, where n=3 and k = 2.5 N/mm, with the points representing the experimental average results (computed as the local slope of torque–angle data in Figure 8) and the solid line representing the theoretical stiffness. (**b**) Same protocol with a stiffer spring (k = 4.3 N/mm). Relative to (**a**), both the stiffness level and its variation with θ increase across all λ.

**Figure 10 micromachines-16-01365-f010:**
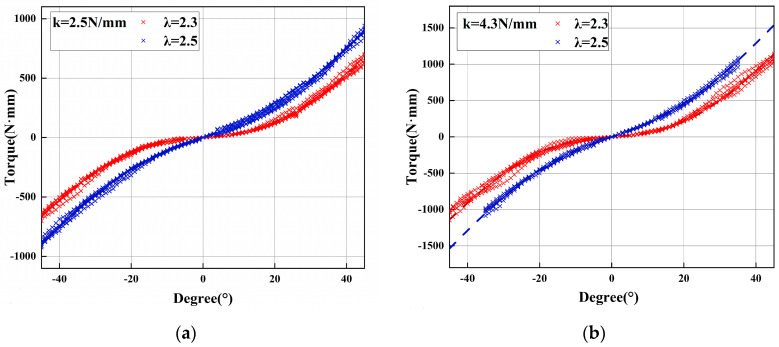
(**a**) Pendulum swing experiment, corresponds to λ = 2.3 and λ = 2.5, each executed for at least six oscillation cycles from −45° to 0° to 45°, with n=3 and k = 2.5 N/mm. (**b**) With a stiffer spring (n=3 and k = 4.3 N/mm). The motion profile is identical to (**a**) (six cycles between −45° and 45°). The torque–angle curves exhibit a steeper slope, and the λ = 2.5 setting yields consistently higher torque than λ = 2.3.

**Table 1 micromachines-16-01365-t001:** Key dynamic parameters of the proposed VSA.

Parameter	Value	Unit
Size (length × width × height)	60×ϕ120	mm
Weight	0.8	kg
Number of spring (n)	Up to 3	pair
Stiffness of spring (k)	2.5/4.3	N/mm
$\mathrm{Length}\;\mathrm{of}\;\mathrm{spring}\;(l_{2.0}$)	30	mm
$\mathrm{Length}\;\mathrm{of}\;\mathrm{input}\;\mathrm{rod}\;(l_{1}$)	24	mm
$\mathrm{Length}\;\mathrm{of}\;\mathrm{output}\;\mathrm{rod}\;(l_{3}$)	12 to 60	mm

**Table 2 micromachines-16-01365-t002:** Spring parameters.

		Design Parameters		Spring Selected	
Spring	Type	Stiffness	Initial Tension	Length at Permissible Load	Model
Spring 1	Linear spring	2.5 N/mm	5.1 N	5.1 mm	MiSuMiC-AWT5–25
Spring 2	Linear spring	4.3 N/mm	9.81 N	7.3 mm	MiSuMiC-AWT8–30

## Data Availability

The data presented in this study are available on request from the corresponding author. The data are not publicly available due to privacy restrictions.

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
