# Peer review of "Design, Modeling, and Testing—A Compact Variable-Stiffness Actuator for Knee Joint Dimensions"

_micromachines, 2025, doi:10.3390/mi16121365_

Round 1

Reviewer 1 Report

Comments and Suggestions for Authors

This paper proposes a mechanism for varying stiffness. After clarifying its characteristics through analytical methods, it experimentally verifies those characteristics. While the results appear meaningful, the explanations are insufficient, and the descriptions lack persuasive detail. I recommend that the authors revise the manuscript thoroughly and consider resubmitting it as a new submission.

  1. Torque transmission seems to occur along the tangential direction of the spring connecting the input and output bars. However, the manuscript does not clearly explain how this aspect is considered in the analysis.
  2. The paper begins with a four-bar linkage model and then transitions to a three-link model. This structural shift causes confusion. Given the configuration of the variable stiffness mechanism—where the input and output bars share the same rotational axis—the three-link model shown in Figure 2 sufficiently explains the mechanism. Therefore, the initial introduction of the four-bar model appears unnecessary and contributes to confusion.
  3. Although the need for a variable stiffness mechanism in rehabilitation is acknowledged, the authors’ intentions remain vague. Section 5 mentions joint stability, collision safety, and energy efficiency, but these aspects are not discussed or supported elsewhere in the manuscript. It is unclear on what basis these claims are made.

  1. I was unable to derive Equation (11) as presented.
  2. The parameters , , and cannot be adjusted after the nuts are tightened, and the nuts consistently lose their fixing torque. This mechanism is not called “tunable”.
  3. The meanings of Figures 8, 9, and 10 are unclear, and they are incorrectly labeled as Figure 2. Some parameters indicate positive stiffness, while others suggest negative stiffness. They are fruitful for the variable stiffness mechanism.
  4. Some references cited in the manuscript could not be found.

I hope these comments are fruitful for the authors.

Author Response

Thank you for the thoughtful and constructive comments. We have revised the manuscript accordingly. For ease of review, we provide a point-by-point response as an attached file. The attachment lists each comment followed by our response and the exact changes made in the manuscript.Please see the attachment.

Reviewer 2 Report

Comments and Suggestions for Authors

The paper is interesting and well written. 

I have only 2 formal observations: 

  1. The references in the paper text are not always shown correctly;
  2. The Figures are not correctly numbered.

Author Response

Thank you for the thoughtful and constructive comments. We have revised the manuscript accordingly. For ease of review, we provide a point-by-point response as an attached file rather than in this text box. The attachment lists each comment followed by our response and the exact changes made in the manuscript.Please see the attachment.

Reviewer 3 Report

Comments and Suggestions for Authors

The authors propose a novel compact variable-stiffness mechanism for a knee exoskeleton intended for stroke rehabilitation. The paper has a good structure, and the topic is highly relevant. However, several aspects could be improved. The following points summarize my recommendations:

  1. Introduction: Several instances of “[Error! Reference source not found.]” appear throughout the section. The authors should revise the reference management to ensure correct citation formatting.

  2. Mechanism structure discussion: Although this is not strictly a correction, I would like to comment that changing the position of the base should not convert the mechanism from a four-bar to a three-bar mechanism. I believe the system remains a four-bar mechanism. The authors should clarify this point.

  3. Line 103: The term ^l1 appears but is not used consistently throughout the article. Please revise for consistency in notation.

  4. Equation (8): I believe this equation is not correct. The authors are encouraged to explain in detail how this equation was derived.

  5. Line 146: The notation N = 3 appears, while Table 1 uses n for the number of springs. If both refer to the same parameter, I recommend using n, since N is commonly used to denote Newtons.

  6. Figure 7: The number 5 (encoder) is difficult to see in the figure. Please improve its visibility.

  7. Table 2: In the column “Permissible load duration,” one row lists units in millimeters (mm) and another in kilograms (kg). These units are incorrect. If the column title refers to a duration, a unit of time should be used.

  8. Figure numbering: After Figure 7, the subsequent figures are labeled as Figure 2. This needs to be corrected.

Finally, the most important aspect that requires clarification is how this mechanism is specifically adapted to patient needs and how it contributes to stroke rehabilitation. Expanding this discussion would significantly strengthen the impact of the work.

Author Response

Thank you for the thoughtful and constructive comments. We have revised the manuscript accordingly. For ease of review, we provide a point-by-point response as an attached file. The attachment lists each comment followed by our response and the exact changes made in the manuscript. Please see the attachment."

Round 2

Reviewer 1 Report

Comments and Suggestions for Authors

The authors have sincerely addressed the reviewers’ comments and revised the manuscript. I consider it suitable for publication. However, I am still curious about the dimension of the middle term in Equation (8), and some equations.

Reviewer 3 Report

Comments and Suggestions for Authors The authors have followed and modified all the suggestions and modifications made by the reviewer.
